# Standoff Detection and Identification of Liquid Chemicals on a Reflective Substrate Using a Wavelength-Tunable Quantum Cascade Laser

**DOI:** 10.3390/s22093172

**Published:** 2022-04-21

**Authors:** Seongjin Park, Jeongwoo Son, Jaeyeon Yu, Jongwon Lee

**Affiliations:** Department of Electrical Engineering, Ulsan National Institute of Science and Technology, Ulsan 44919, Korea; psjin62@unist.ac.kr (S.P.); jeongwoo@unist.ac.kr (J.S.); ujy0314@unist.ac.kr (J.Y.)

**Keywords:** chemical detection, quantum cascade laser, mid-IR laser absorption spectroscopy, standoff detection, diffuse reflectance spectroscopy

## Abstract

Standoff chemical detection and identification techniques are necessary for ensuring safe exposure to dangerous substances. Molecular fingerprints of unknown chemicals can be measured using wavelength-tunable quantum cascade lasers operating in long-wavelength infrared. In this work, we present a method that can identify liquid chemicals on a reflective substrate via diffuse reflection spectra measurement from 50 cm away and multiple nonlinear regression analysis. Experimental measurements and numerical analyses were conducted for different chemical surface densities and angles of light incidence using diethyl phthalate (DEP) and dimethyl methylphosphonate (DMMP). Candidate substances can be classified using a deep learning model to reduce analysis time.

## 1. Introduction

Spectroscopic methods capable of detection and identification in standoff are essential for ensuring safety in situations where dangerous chemical analytes such as chemical warfare agents (CWAs) or explosive residues may be present [1,2]. Laser absorption spectroscopy (LAS) is a promising method for standoff chemical detection and identification with the advantages of high sensitivity, spectral selectivity and resolution, and fast scan speed compared to other conventional methods using non-laser sources such as differential optical absorption spectroscopy or Fourier transform infrared spectroscopy [3]. Various methodologies such as differential absorption LiDAR, wavelength tunable laser absorption spectroscopy, laser photo-acoustic spectroscopy, dual-comb spectroscopy, laser heterodyne radiometry, and active coherent laser absorption spectroscopy have been proposed for LAS and their technical details are summarized in [3]. Wavelength tunable quantum cascade lasers (QCLs) have been developed as coherent light sources that operate in the long-wavelength infrared (LWIR) region typically defined as the 8–14 μm wavelength range, where molecular fingerprints are concentrated due to the intrinsic vibrational absorption of molecules. QCL-based LAS using such molecular fingerprints in the LWIR region provides a useful spectroscopic technique for detecting and identifying chemical substances in gaseous or condensed phases with low atmospheric absorption. Recently, attempts have been made to detect and identify various types of chemicals, such as CWAs [4], volatile organic compounds [5], proteins [6], trace particles [7], and microorganisms [8], using an external cavity QCL (EC-QCL) capable of broadband wavelength tuning. Recently, hyperspectral imaging techniques using EC-QCLs equipped with electromechanical scanners have been reported [7,9]. For gas detection based on QCL, relatively accurate absorption spectrum analysis is possible using an external retroreflector or multipass gas cell. However, liquid-phase chemicals on substrates can hinder accurate detection and identification due to diffusion onto the substrate, evaporation, and spreading across the surface [10].

In this work, we experimentally demonstrate a standoff detection and identification spectroscopic method for two different liquid chemicals on reflective substrates with different surface roughness using a detector system with an integrated wavelength-tunable EC-QCL and HgCdTe (MCT) photodetector. For analysis of the chemicals on the metallic plates, we measured only diffuse reflection spectra using a slight off-normal incidence of light and a wavelength scan of the EC-QCL. Diffuse reflection has a relatively low signal intensity compared to specular reflection but hardly contains any Fresnel reflection components that appear by refractive index contrast on the surface of a liquid chemical and which can greatly relax alignment conditions during measurement. More importantly, diffuse reflection contains mostly absorption spectra information of liquid chemicals because incident light propagates inside the liquid chemical and is scattered by the surface [11]. In general, diffuse reflection spectra need to be normalized to the reflection spectra of a bare substrate containing the analyte. This normalization requires additional measurements for the extraction of absorption spectra only. However, the standoff detection of liquid chemicals on substrates involves varying background reflection spectra depending on the substrate type, making extracting the exact absorption spectra every time very challenging. Therefore, we developed an algorithm that matches the measured diffuse reflection spectra with the calculated spectra by fitting the values of a theoretical model using the least square method, enabling detection and identification without background reflection measurements. Then, we experimentally validated our theoretical model and built a deep learning classification model to identify analyte chemicals using a data set calculated from the model. Our method can be used for the detection and identification of unknown substances on a rough reflective substrate.

## 2. Detector Configuration

The detector used in this study consists of a wavelength-tunable EC-QCL (Daylight Solutions, Inc., tuning range: 930–1190 cm^−1^, spectral resolution: 2 cm^−1^, beam divergence: <4 m rad), a laser controller, an MCT photodetector, two parabolic mirrors, a guide visible laser, a signal amplifier, a data acquisition system (DAQ), and a miniPC, as shown in Figure 1a. An image of the detector is shown in Figure 1b. The coaxially aligned visible and LWIR light from the sources pass through the center hole of the parabolic mirror and an anti-reflection coated ZnSe window, and incident with an angle *θ* onto a substrate. The laser spot diameter at the substrate 50 cm away from the detector was about 3 mm. Part of the light backscattered (diffuse reflection) toward the detector is received and focused onto the MCT detector by the two parabolic mirrors. The analog signal from the MCT photodetector is integrated by the signal amplifier, and then the integrated analog signal is converted to a digital signal from the DAQ. Finally, the digital signal is transferred to the miniPC attached to the side of the detector. Figure 1c shows the averaged power spectra of the QCL with a 5% duty measured at the location of the substrate using a calibrated optical power meter. Due to the low power at the two spectral edges of the QCL, a wavenumber tuning range of 950–1174 cm^−1^ was used for the QCL.

## 3. Experimental Configuration

### 3.1. Reflective Substrates

Diffuse reflection is generally affected by surface roughness. Roughness-standard nickel plates with six different surface roughness values (Ra in μm: 0.05, 0.1, 0.2, 0.4, 0.8, and 1.6) were used for diffuse reflection spectra analysis of liquid chemicals on reflective substrates. A photograph of the substrate is shown in Figure 2a. The distance between the detector and the substrate was fixed at 50 cm. Figure 2b shows the backscattering signals measured by the MCT when QCL light with a wavenumber of 1090 cm^−1^ is irradiated onto the substrates with different surface roughnesses at different incidence angles. In the case of a smooth surface with low roughness, the backscattering signal at off-normal incidence is low because specular reflection is dominant. Random back scattering is severe at an Ra of 1.6 μm due to the high roughness, and the MCT signal becomes lower than that of Ra 0.8 μm. Although the maximum backscattering signal intensity changes according to the surface roughness and angle of incidence, it has been experimentally proven that the spectral shape of the backscattering signal generated from the reflective surface is almost constant regardless of the roughness and incidence angle. Figure 2c shows the normalized reflection power spectra extracted using the average value of the reflection spectra measured in all the cases above. This single power spectrum was used for diffuse reflection spectral analysis without measuring the background spectra.

### 3.2. Analyte Chemicals

Experiments were conducted using diethyl phthalate (DEP), a simulant of VX, and dimethyl methylphosphonate (DMMP), a simulant of G-agent [12]. Figure 3a,b show the normalized absorbance spectra of DEP and DMMP, respectively, obtained from the National Institute of Standards and Technology Chemistry WebBook [13]. The absorbance spectra were cubic interpolated to match the actual wavenumbers of the QCL tuning range [5]. Ethanol was used as the solvent to control the surface density of the analyte. The analyte chemicals were distributed on the substrates with five different surface densities (1, 2, 3, 4, and 5 g/m^2^) using a two-dimensional translation stage and a syringe pump to control the flow rate for uniform coating. However, spatial variation in the surface density was unavoidable due to the chemical viscosity and surface tension. Figure 3c shows photographs of DEP distributed on the substrates formed by grinding for a surface density of 4 g/m^2^. All measurements were conducted 2 min after coating so that the ethanol was sufficiently evaporated and only the analyte chemicals would be present on the surface at the target surface density.

## 4. Spectral Analysis and Identification

### 4.1. Multiple Nonlinear Regression

We used the multiple nonlinear regression (MNR) method for spectral analysis of the measured data and identification of analyte chemicals [5,14]. Based on the Beer–Lambert law, we can express the experimental diffuse reflection spectra (*Y*(*λ*)) using Equation (1).
(1)$$Y(\lambda )=c_{0}L(\lambda )\prod_{i}\exp\left(c_{i}\alpha_{i}(\lambda )\right)+\varepsilon (\lambda )$$
where *L*(*λ*) is the normalized QCL power spectra shown in Figure 2c; *c*_0_ is the power scaling factor for fitting to actual measurement spectra; *c_i_* and *α_i_*(*λ*) are the absorbance scaling factor and the normalized absorbance spectra of the *i*th analyte chemical, respectively; and *ε*(*λ*) is the residual caused by thermal noise, external reflection, and background mismatch. The power scaling factor *c*_0_ is affected by the properties of the substrate and the light incidence angle. It is used for fitting with the experimental value by adjusting the intensity of the diffuse reflection signal. The absorbance scaling factor *c_i_* is proportional to the actual path length of the light through the analyte at a specific incidence angle. The scaling factors are obtained by iteration based on the least square method (LSM), and the minimum of the mean squared residual over the wavenumber range of 950 to 1174 cm^−1^ is determined. The coefficient of determination *R*^2^ represents how closely the measured spectra fit the modeled spectra and is expressed by Equation (2) [15]:(2)$$R^{2}=1-\frac{\sum {\left(\varepsilon (\lambda )\right)}^{2}}{\sum {\left(Y(\lambda )-\bar{Y}\right)}^{2}}$$
where $\bar{Y}$ is a single averaged value of *Y*(*λ*) in the wavenumber range of 950–1174 cm^−1^. In calculating *R*^2^, the measured spectra were preprocessed using the Savitzky–Golay smoothing filter. *R*^2^ has a value between 0 and 1, with a higher score corresponding to a better match between the two spectra.

### 4.2. Deep Learning Classification

In the detection and identification of unknown chemicals, it is time consuming and inefficient to compare the spectral data of all chemicals stored in the library and the measured values. If the type of candidate chemical can first be determined through the measured spectral data, the time required for calculation can be reduced, and the accuracy for identification can also be increased. Accordingly, the deep learning classification method can be utilized to determine the types of candidate chemicals [16]. Building a classification model requires a large amount of training data and collecting enough data from actual measurements is time consuming. We generated a large amount of data for machine learning without actual measurement by assigning random values to the fitting parameters *c*_0_ and *c_i_*. In this work, we created s calculated data set for four data classes: “None”, “DEP”, “DMMP”, and “Mixture of DEP and DMMP”. The dataset was used to train the classification model using the neural networks of the MATLAB program. The minimum value of *c*_0_ is 0.03, which filters out very low signals that cannot be detected and identified. *c_i_* has a value between 0.1 and 10, which was determined experimentally. By applying the classification model to the experimental results, it was verified that the candidate group was well classified.

## 5. Experiments

### 5.1. DEP Experiment

The analytical model for DEP detection can be expressed as using the DEP absorbance α_DEP_ and its absorbance scaling factor, *c*_DEP_, as $Y(\lambda )=c_{0}L(\lambda )\exp\left(c_{\mathrm{DEP}}\alpha_{\mathrm{DEP}}(\lambda )\right)+\varepsilon (\lambda )$ using Equation (1). The coefficient of determination *R*^2^ was derived from the measured spectra and the analytical model to which scaling factors, extracted through iteration, were applied. We used a reflective substrate with an Ra of 1.6 μm. The reflective substrate coated with DEP was scanned at incidence angles of 2, 5, 10, 15, 20, and 30° at 50 cm away from the detector. Figure 4a shows an example of the measured and calculated diffuse reflection spectra for a surface density of 2 g/m^2^ at an incidence angle of 5°. The calculated diffuse reflection spectra were obtained using Equation (1) without the residual *ε*(*λ*). In this case the *c*_0_, *c*_DEP_, and *R*^2^ values extracted by the LSM were 0.87, 0.23, and 0.99, respectively, indicating good agreement between the experiment and calculation. Figure 4b shows the measured and calculated diffuse reflection spectra for a surface density of 4 g/m^2^ at an incidence angle of 5°. The corresponding *c*_0_, *c*_DEP_, and *R*^2^ values extracted by the LSM were 0.27, 1.49, and 0.90, respectively. In this case, a larger amount of DEP liquid was coated on the surface, which caused more specular reflection on the smooth surface of the chemical, lowering the diffuse reflection signal measured by the MCT. The power scaling factor *c*_0_ extraction results with regard to the surface density and incidence angle are shown in Figure 4c. The intensity of the diffuse reflection signal, which was proportional to the power scaling factor *c*_0_, tended to decrease as the surface density and incidence angle increased. Figure 4d shows the DEP absorbance scaling factor *c*_DEP_ extracted from the measured data for different surface densities and incidence angles. *c*_DEP_ tended to increase as the surface density increased, but decreased at a surface density of 5 g/m^2^, as compared to that at 4 g/m^2^. Presumably, a uniform film was not formed on the substrate at a high surface density, and the diffuse reflection portion was relatively reduced, thereby reducing the absorbance scaling factor. Further, as the incidence angle increases, the effective path length increases, and *c*_DEP_ also shows a general tendency to increase. In addition, as described above, a slight deviation from this trend is noted at high surface densities. Figure 4e shows the distributions of the coefficient of determination *R*^2^ obtained from the measured data and the analytic model according to *c*_0_. When *c*_0_ is higher than 0.03, most of the *R*^2^ values exceed 0.9. Contrarily, at *c*_0_ < 0.03, most of the *R*^2^ values are distributed below 0.9. Therefore, to identify DEPs with more than 90% accuracy, the threshold value of *c*_0_ can be set to 0.03. As shown in Figure 4c, a *c*_0_ value of 0.03 or higher can be obtained at incidence angles in the range of 2°–30° and at surface densities in the range of 1–5 g/m^2^. Thus, the chemical can be analyzed over a wide concentration range with minimal device alignment. Next, we used data with a *c*_0_ value above the threshold and checked whether classification was well performed using the deep learning model. In this process, the experimental data obtained using all the substrates (flat lapping, reaming, grinding, horizontal milling, vertical milling, and turning) in Figure 2a were used. Figure 4f shows the classification results of the pretrained deep learning model. Out of the total 363 experimental spectral datapoints, 290 cases were correctly classified as DEP with a classification accuracy of 80%. The number of results incorrectly classified as None, DMMP, and Mixture were 25, 8, and 40, respectively. The results classified as Mixture can be corrected during the MNR fitting method. For example, the measured data in Figure 4c was classified as Mixture, but when MNR with the theoretical model of Mixture was applied, a negligibly low DMMP absorbance scaling factor of 1.04 × 10^−7^ was obtained; for comparison, the obtained DEP absorbance scaling factor was 1.8. Therefore, the final identification accuracy after the MNR fitting was 90.9%, and the accuracy could be further improved if training was performed using a dataset extracted under more diverse test conditions.

### 5.2. DEP and DMMP Mixture Experiment

To verify whether multiple chemicals can be detected and identified through diffuse reflection measurements and the MNR method, additional tests were conducted using a mixture of DEP and DMMP with a 1:1 ratio coated on the reflective substrate and an Ra of 1.6 μm. The surface density used in this experiment was 5 g/m^2^. The diffuse reflection spectra of the mixture of DEP and DMMP can be modeled using Equation (1) as $Y(\lambda )=c_{0}L(\lambda )\exp\left(c_{\mathrm{DEP}}\alpha_{\mathrm{DEP}}(\lambda )+c_{\mathrm{DMMP}}\alpha_{\mathrm{DMMP}}(\lambda )\right)+\varepsilon (\lambda )$ where *α*_DMMP_ and *c*_DMMP_ are the normalized DMMP absorbance (cf. Figure 3b) and its absorbance scaling factor, respectively. Figure 5a shows the measured and calculated diffuse reflection spectra of the DEP and DMMP mixtures at an incidence angle of 5°. The calculated diffuse reflection spectra were obtained using Equation (1) without the residual *ε*(*λ*). The *c*_0_, *c*_DEP_, *c*_DMMP_, and *R*^2^ extracted by LSM were 0.22, 0.60, 0.66, and 0.94, respectively. Figure 5b shows the absorbance spectra of DEP, DMMP, and the mixture calculated using the extracted absorbance scaling factors. The calculated reflection spectra in Figure 5a, where reflection valleys occur at the main absorption peaks of the mixture, can be obtained by considering the laser power spectra. Figure 5c shows the absorbance scaling factors of DEP (*y*-axis) and DMMP (*x*-axis) extracted from measured data using the substrate with different Ra values at an incidence angle of 5°. The two absorbance scaling factors varied according to the surface roughness of the substrate, but the ratio of the two values remained constant. For the substrate with fixed surface roughness of Ra = 1.6 μm, it was confirmed that the absorbance scaling factors increased as the incidence angle increased due to the increased effective path length as shown in Figure 5d, which showed the same trend as the single DEP measurement (cf. Figure 4d). Finally, we used the mixture measurement data with a *c*_0_ value higher than 0.03 and checked whether classification was well performed through the deep learning model. As shown in Figure 5e, all 20 of the measured data were correctly classified to Mixture.

## 6. Conclusions

In this work, we demonstrate an identification method for liquid chemicals on a reflective substrate based on diffuse reflection measurement and MNR. The measured spectra and the analytical model were fitted using a power scaling factor, absorbance scaling factors, and a normalized background power spectrum. An *R*^2^ of 0.9 or higher was obtained from data with a power scaling factor of 0.03 or higher, which could be measured in the surface density range of 1–5 g/m^2^ and an incidence angle range of 2°–30°. These results indicate that liquid chemicals can be identified with very high accuracy under very relaxed detector alignment conditions without background spectrum measurement in environments where analysis through diffuse reflection is possible, such as analyte chemicals on a rough reflective substrate. In addition, through the deep learning model, candidate groups can be extracted with high accuracy without having to directly compare all libraries stored inside the detector, which can drastically reduce the computation time for the identification of unknown chemicals.

## Figures and Tables

**Figure 1 sensors-22-03172-f001:**
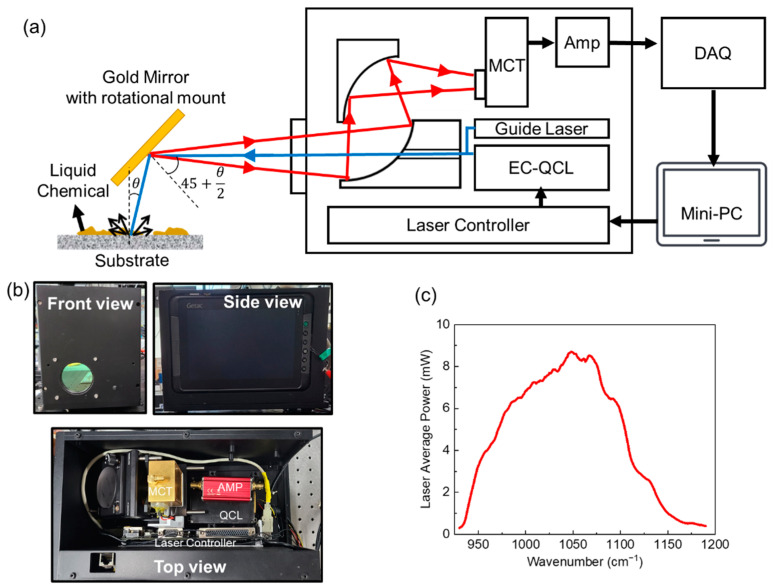
(**a**) Schematic of the detector with block diagram and measurement configuration. The blue and red arrow indicate incident light from the lasers and reflected light from the target substrate, respectively. (**b**) Front (top left), side (top right), and top view (bottom) images of the detector. (**c**) Averaged QCL power spectra measured at the position of the target substrate.

**Figure 2 sensors-22-03172-f002:**
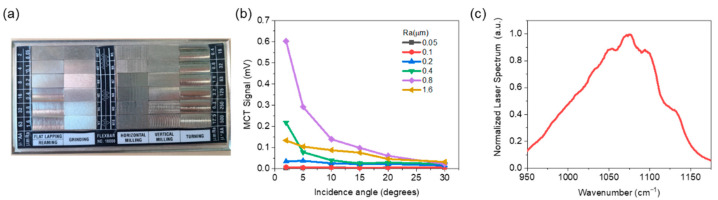
(**a**) Photograph image of the roughness standard nickel plates. (**b**) Backscattering signal measured by the MCT when the QCL light at a wavenumber of 1090 cm^−1^ is irradiated onto the reflective substrates with different surface roughnesses at different incidence angles. (**c**) Normalized QCL power spectra averaged over all measured values using reflective substrate with different surface roughness and different incidence angle.

**Figure 3 sensors-22-03172-f003:**
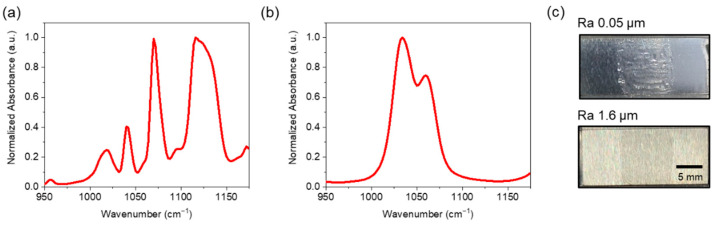
Normalized absorbance spectra of the analyte chemicals of (**a**) DEP and (**b**) DMMP in the EC-QCL operational wavenumber range. (**c**) Photograph images of DEP with surface density of 4 g/m^2^ distributed on the substrates formed by grinding.

**Figure 4 sensors-22-03172-f004:**
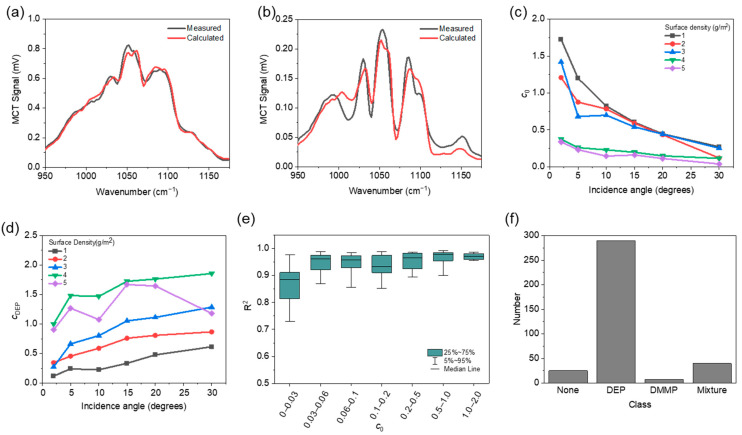
(**a**,**b**) Measured (black) and calculated (red) diffuse reflection spectra for DEP coated substrate with Ra of 1.6 μm, at incidence angle of 5° for surface density of (**a**) 2 g/m^2^ and (**b**) 4 g/m^2^. (**c**,**d**) Extracted power scaling factor (**c**) *c*_0_ and extracted absorbance scaling factor (**d**) *c*_DEP_ from measured data for different surface densities of DEP and different incidence angles using the substrate with Ra of 1.6 μm. (**e**) Distribution of the coefficient of determination *R*^2^ according to *c*_0_. (**f**) Result of None, DEP, DMMP, and Mixture of DEP and DMMP.

**Figure 5 sensors-22-03172-f005:**
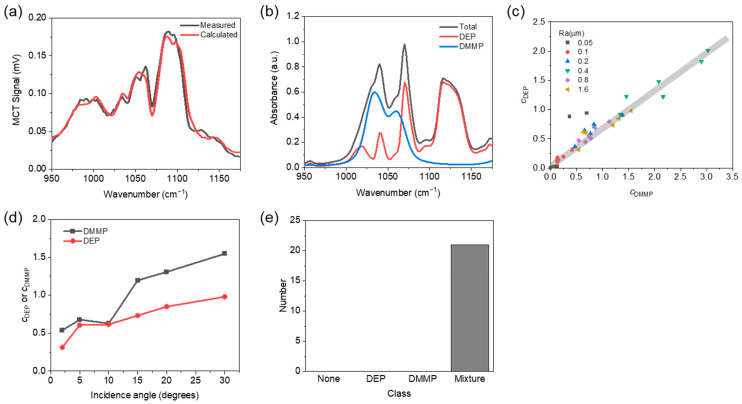
(**a**) Measured (black) and calculated (red) diffuse reflection spectra of DEP and DMMP 1:1 ratio mixture coated on substrate with Ra of 1.6 μm at incidence angle of 5° for surface density of 5 g/m^2^. (**b**) Calculated absorbance spectra of DEP (red), DMMP (blue), and 1:1 ratio mixture (black) using the extracted absorbance scaling factors in (**a**). (**c**) Absorbance scaling factors of DEP (*y*-axis) and DMMP (*x*-axis) for a surface density of 5 g/m^2^ and different surface roughness of the substrate. (**d**) Absorbance scaling factors of DEP and DMMP for substrate roughness Ra of 1.6 μm, surface density of 5 g/m^2^ according to incidence angle. (**e**) Result of classification of the mixture measurement data using the deep learning model trained for four classes of None, DEP, DMMP, and Mixture of DEP and DMMP.

## Data Availability

Data underlying the results presented in this paper are available from the authors upon request.

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
