# Peer review of "Standoff Detection and Identification of Liquid Chemicals on a Reflective Substrate Using a Wavelength-Tunable Quantum Cascade Laser"

_sensors, 2022, doi:10.3390/s22093172_

Round 1

Reviewer 1 Report

The authors studied standoff detection of liquid chemicals on a reflective substrate using a wavelength-tunable quantum cascade laser. They identified liquid chemicals via diffuse reflection spectra measurement and multiple nonlinear regression. It sounds interesting and is possibly to be a promising application of tunable laser absorption spectroscopy.

1) The atmospheric absorption in the target spectral band, i.e. potential atmospheric interferences, should be considered and analysis in details for standoff detection;

2) It is common to use absorbance spectra from recognized database for infrared spectroscopy, one should pay attention to their spectral resolution, and denote how they match the spectral resolution between the experimental and the references.

3) Recently, standoff detection is a rapidly developing technical fields, a comprehensive review in the introduction is suggested.

Reviewer 2 Report

The manuscript by Park et al. reported a method for standoff identification of liquid chemicals on a reflective substrate via diffuse reflection spectra measurement. These results can be interesting for readers. However, several minor issues must be clarified to improve the paper's quality. 

1) How were the calculated reflection spectra obtained?

2) There are several questions regarding the experimental part:

  1. How the variation of the incident angle θ was realized?
  2. What is the laser beam divergence and what is the size of laser spot at substrate?
  3. How was a surface roughness of test nickel plates measured?
  4. What is about the measurement error in Fig. 2b?
  5. Is it possible to use a spin-coating to decrease spatial variations in the surface density of analyte?
  6. For classification results presented in fig. 4f (DEP experiment) the total number of tests is 363. But in fig. 5e (DEP and DMMP mixture experiment) the total number is only 20. Therefore, sample sizes are quite different.

3) Is it possible to use the proposed method in the case the distance between the detector and the substrate is not 50 cm? The same question is for the cases if the surface density is out of range of 1–5 g/m2 and an incidence angle is out range of 2–30°.  

4) Some minor problems are found through the text:

  • The scale bar is missed in fig. 3c.
  • It seems that equations (3) and (4) are special cases of equation (1) and therefore can be eliminated.

Round 2

Reviewer 1 Report

Agree to be accept.